biochemistry/cellular biology/molecular biology

PyMDI-Zn, protein quantitation, SDS-PAGE, live cell imaging

**Author for correspondence:**
Zhuo Tang
e-mail: tangzhuo@cib.ac.cn

# A general fluorescent light-up probe for staining and quantifying protein

Jiawei Zou[1,2], Gangyi Chen[1], Feng Du[1], Yi Yuan[1], Xin Huang[1], Juan Dong[1], Kexin Xie[1], Xin Cui[1] and Zhuo Tang[1]

[1]Natural Products Research Center, Chengdu Institute of Biology, Chinese Academy of Sciences, Chengdu, Sichuan 610041, People's Republic of China
[2]University of Chinese Academy of Sciences, Beijing 100049, People's Republic of China

ZT, 0000-0002-5845-3569

Proteins are the primary functional agents in all cellular processes, facilitating various functions such as enzymes and structure-forming or signal-transducing molecules. In this work, we report a fluorescent dye, PyMDI-Zn, which could specifically bind with proteins and provide a red-shifted fluorescent emission. The visual analysis of protein in sodium dodecyl sulfate-polyacrylamide gel electrophoresis could be realized in 5 min by using PyMDI-Zn as a light-up dye. Based on its cell penetration and low toxicity, PyMDI-Zn could also be applied to locate protein-rich regions and organelles in live cell imaging. Moreover, the direct protein quantitation can be realized based on PyMDI-Zn, providing a method of screening for food adulteration by nitrogen-rich compounds.

## 1. Introduction

Proteins are essential players in most biological systems and are involved in a variety of complicated biological processes including signal regulation, small molecule transport, immunity and enzyme catalysis. [1,2]. Protein analysis with rapid response, high sensitivity and easy handling is of fundamental importance for understanding the diverse functions of proteins. Several methods have been developed for both qualitative and quantitative detection of protein: (1) absorption spectrometry, (2) colorimetry, (3) fluorescence spectrometry. Absorption spectrometry is based on measuring the ultraviolet absorbance of aromatic amino acids in a protein at different wavelengths [3,4]. The colorimetric determination of protein concentration uses the chromophores that can bind with protein and exhibit a colour change. Bradford assay is the most widely used colorimetric method for the detection of protein in solution or gel for electrophoresis by using

Coomassie Brilliant Blue (CBB) as the protein-binding dye. However, this assay exhibits various responses for different proteins and most commercial detergents could interfere with the protein assay [5].

Fluorescent spectrometry is of particular interest for the staining and visualizing of proteins because of its high sensitivity and convenience. Several fluorescent reagents targeting proteins have been developed and widely applied for the detection of proteins in solution or gel for electrophoresis, e.g. Sypro Ruby, Cyanines, BODIPY, pyridinium bromide and Fluorescamine. However, disadvantages, such as insolubility in an aqueous phase, aggregation of dyes and instability under ambient conditions, impose restrictions on their practical application [6]. And the response of some fluorescent reagents, such as Sypro Ruby, against proteins, will be greatly affected by detergents like sodium dodecyl sulfate (SDS), thus it takes a long time to get rid of SDS for staining proteins in SDS-polyacrylamide gel electrophoresis (SDS-PAGE) [7]. Besides, fluorescent dyes, such as Alexa Fluor, BODIPY and Sypro Ruby, have also been widely used in cell imaging to provide critical insight into the basic nature of cellular function, while they must be modified with a specific probe for a certain protein, such as phalloidin conjugated to Alexa Fluor for F-actin, glibenclamide conjugated to BODIPY for endoplasmic reticulum, ceramide conjugated to BODIPY for Golgi complex and so on [8–15]. However, these fluorescent dyes designed for fluorescence imaging of tissue distribution and subcellular localization have some significant disadvantages which include (i) requiring of cell fixation, (ii) more or less cytotoxicity, (iii) high cost due to complex chemical synthesis.

Therefore, to achieve high efficiency for protein detection *in vitro* or *in vivo*, novel fluorescent probes with high specificity, resistance to interference from foreign substances, ease of use and broad dynamic range are highly desired [16–19]. Herein, we report an interesting compound, named (Z)-1,2-dimethyl-4-(pyridin-2-ylmethylene)-1H-imidazol-5(4H)-one (PyMDI), which could bind with proteins specifically in the presence of $Zn^{2+}$ ions as a light-up probe through emitting strong green fluorescence. PyMDI-Zn revealed a fast response to proteins and high resistance to interference from foreign substances. Different protein-targeting detection methods, including the fast and simple fluorescent staining in SDS-PAGE, live cell imaging and accurate quantitation in solution, have been realized based on this new fluorescent molecular probe.

# 2. Material and methods

## 2.1. Materials

Bovine serum albumin (BSA), human serum albumin (HSA), Brilliant blue R–250, Pyronin Y, DAPI and SYPRO (R) Orange Protein Gel Stain were purchased from Sigma Chemicals Co. All other chemicals are of analytical reagent grade. JM109, DH10B, BL21(DE3) Chemically Competent Cell, ProteinRuler I and ProteinRuler II, were purchased from TransGen Biotech (Beijing, China). A Pierce Rapid Gold BCA Protein Assay Kit was purchased from Thermo Fisher Scientific. RecR, RuvA, RuvB, RecA and MutL were cloned into an expression vector, and the proteins were purified. Cell cultures were maintained in the DMEM medium supplemented with 10% heat-inactivated fetal calf serum, 1% penicillin/streptomycin storage solution (100 U/ml penicillin, 100 mg/ml streptomycin), at 37°C in a humidified atmosphere containing 5% $CO_2$. All the solutions were prepared with sterile ultrapure water. The (Z)-1,2-dimethyl-4-(pyridin-2-ylmethylene)-1H-imidazol-5(4H)-one (PyMDI) was synthesized according to the synthesis procedure described in [20].

## 2.2. Instruments

Absorbance, excitation and emission spectra were measured with a Thermo Scientific Varioskan Flash. Flow cytometry experiments were performed on a MoFlo™ XDP Cell Sorter (Beckman Coulter). Cell images were recorded using a Leica TCS SP8 Confocal Microscope. Laser 355 and 488 nm were selected when using a flow cytometer, and laser 405, 488 and 552 nm for Confocal Microscope. SDS-PAGE was performed on the Invitrogen SureCast™ system, and the gel images were recorded by GE Healthcare Bio-Sciences AB Typhoon FLA 7000 and Canon Digital camera EOS 600D. The Kjeldahl method was performed on a Foss Kjeltec 2200.

## 2.3. Electrophoresis and staining with PyMDI

Proteins were loaded in 12% SDS-PAGE. The stacking layer runs at 40 V, and the resolving layer runs at 80 V. The gel was rinsed with deionized water and stained with PyMDI-Zn (100 µM) for 5 min at room

temperature (25°C). Coomassie stain protocol: the SDS-PAGE gel was rinsed with deionized water, and then stained with Coomassie solution, incubation with gentle agitation at room temperature for 1–20 h. Gels were destained overnight with gentle agitation.

## 2.4. Cellular location

Hela and HepG2 cells were grown in a 20 mm glass-bottom cell culture dish, and the cells were washed with PBS three times, followed by incubation with PyMDI-Zn (50 µM), DAPI (nucleus location because of binding to DNAs) and Pyronin Y (nucleolus location because of binding to RNAs) for 30 min at 37°C. And samples were viewed under a Leica TCS SP8 Confocal Microscope. Images were obtained using the LAS AF software, then subsequently processed with the Adobe Photoshop program.

## 2.5. Cell viability assay

HepG2 was cultured in a 96-well plate, the cells were exposed to PyMDI-Zn at a concentration in the range of $50 \sim 500 \, \mu M$ (DMSO as a control) for 24 h, then the medium was replaced with 200 µl DMEM containing 10% (v/v) Alamar Blue. After 2 or 3 h, fluorescence was monitored at wavelength 530 nm for excitation and 590 nm for emission by Thermo Scientific Varioskan Flash.

## 2.6. Determination of protein concentration

One hundred microlitres of solutions containing different amounts of protein (0–600 µg) with 100 µM PyMDI-Zn were incubated at room temperature for about 5 min. Then, the fluorescence of those samples was detected at 520 nm with excitation at 486 nm by Thermo Scientific Varioskan Flash. The concentration of protein was plotted against the corresponding fluorescent intensity to obtain a standard curve. The protein concentration of the unknown samples was determined by the standard curve. Protein solutions are normally assayed in triplicate.

# 3. Results and discussion

## 3.1. Spectral behaviour of PyMDI

The green fluorescent protein-like chromophore (PyMDI) has been developed and carefully investigated by the Tolbert group. According to their results, PyMDI would emit blue fluorescent light in the presence of $Zn^{2+}$ ions with the absorption/emission maxima (figure 1*a*) [20]. When we applied PyMDI-Zn to stain *Escherichia coli* cells, however, very weak fluorescent signals could be detected with a flow cytometry or confocal microscope with 355 nm laser radiation (figure 1*b* and electronic supplementary material, figure S1). Unexpectedly, the flow cytometry received a strong fluorescent emission at 520 nm with excitation at 488 nm (figure 1*b*), and cell imaging with a confocal microscope showed similar results (electronic supplementary material, figure S1). Hence, these unexpected results triggered us to carry out more experiments to investigate the property of PyMDI-Zn. The green fluorescence of PyMDI-Zn might be induced by some components, and some probable biomolecules (DNA, RNA, protein, sugar or polysaccharide) were carefully investigated *in vitro*. As illustrated in figure 1*c*, the fluorescent intensity of PyMDI-Zn increased about six times at 520 nm with excitation at 488 nm in the presence of BSA, while there was no increased signal that could be detected using other samples (figure 1*c*). According to excitation/emission spectra (figure 1*d*), we found that PyMDI-Zn–BSA complex had an excitation maximum at 486 nm (blue dashed line) and an emission maximum at 520 nm (green solid line). Together, PyMDI-Zn could light up (520 nm) when binding to proteins and excited around 486 nm, indicating that PyMDI-Zn might be a promising tool for protein detection and analysis.

## 3.2. Fast protein detection by SDS-PAGE

To test the application of PyMDI-Zn in protein analysis, we first applied PyMDI-Zn for protein staining in SDS-PAGE. As illustrated in figure 2*a*, nine proteins of different sizes (22–94 kDa) were stained with PyMDI-Zn. All proteins in SDS-PAGE could be successfully stained with 100 µM PyMDI-Zn, and the bands corresponding to different proteins could be seen clearly under UV irradiation with or without

**Figure 1.** Spectral behaviour of PyMDI-Zn (*a*) Absorption (ab) and emission (em) spectra of PyMDI. (*b*) *E. coli* cells stained with PyMDI-Zn were performed on MoFlo™ XDP Cell Sorter (Beckman Coulter), including *E. coli* cells without staining (black), *E. coli* cells stained with PyMDI-Zn, excited at 355 nm and detected at channel FL10 (457/50) (blue) and 488 nm for channel FL1 (529/28) (green). (*c*) Fluorescent intensity was collected in the presence of excess DNA, RNA, glucose, glycogen, starch and BSA mixed with PyMDI-Zn, PyMDI-Zn for background control. (*d*) Excitation spectrum (blue dashed line) and emission spectrum (green solid line) of PyMDI-Zn–BSA complex.

washing step (electronic supplementary material, figure S2), which is comparable to the staining result with CBB. In general, foreign substances, such as detergents, chelating reagents, inorganic salts and organic solvents, could significantly change the response of dye to proteins, thus it usually takes a long time to get rid of them for proteins staining in SDS-PAGE. However, the washing step is no longer needed when using PyMDI-Zn to stain protein in our experiment, implying that this new probe may be more tolerant of interference from other non-protein substances. We found that the fluorescence intensity of PyMDI-Zn–BSA is very stable in most detergents and retains 90% fluorescence at the concentration of 10% for SDS, 10% for Triton X-100, 2.5% for NP-40, 0.25% for Tween 20, respectively. Then we selected more potentially interfering compounds including buffering agents, chelating reagents and organic solvents. The responses of PyMDI-Zn to these commonly used reagents were investigated, revealing that the protein detection methods based on PyMDI-Zn probe could tolerate most interfering compounds which could be frequently encountered during the purification and detection of proteins (electronic supplementary material, table S1 and figure S3).

Usually, Coomassie blue staining is the first choice for protein detection in SDS-PAGE for most researchers. Unfortunately, the CBB staining method undergoes a complex process, in which protein in SDS-PAGE needs to be fixed and stained for 3 h to overnight, and then rinsed in deionized water or dilute methanol/acetic acid solution for at least 30 min [21–24]. In our experiments, by using PyMDI-Zn as the probe to stain proteins in SDS-PAGE, it only took 5 min to achieve half maximum staining, and the maximum intensity could be obtained in 10 min, which is comparable to the staining result with CBB (figure 2*b*). It also shows that our approach could provide good sensitivity, a contrast to CBB. Moreover, the total proteins of *E. coli* bacteria were separated by 12% SDS-PAGE and stained with 100 μM PyMDI-Zn for 5 min. We obtained the same staining result as that of CBB method (figure 2*c*), which is comparable to the commercially available fluorescent probe for protein gel stain (electronic supplementary material, figure S4), revealing that PyMDI-Zn could respond to proteins

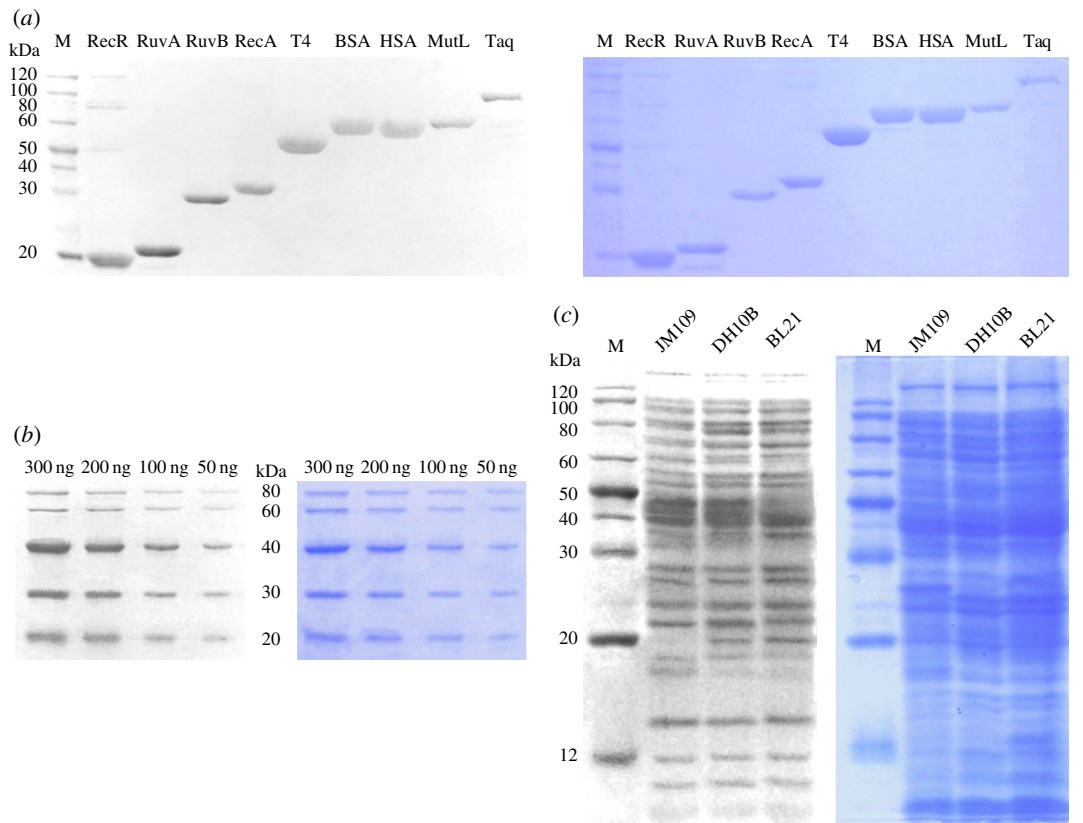

**Figure 2.** Protein staining in SDS-PAGE with PyMDI-Zn. (*a*) Nine proteins were separated by 12% SDS-PAGE, RecR (22 kDa); RuvA (24 kDa); RuvB (38 kDa); RecA (39 kDa); T4 DNA ligase (55 kDa); BSA (66 kDa); HSA (66 kDa); MutL (69 kDa); Taq polymerase (94 kDa). (*b*) Protein samples were commercial protein markers, which contain proteins of 80 kDa (100 ng μl$^{-1}$), 60 kDa (100 ng μl$^{-1}$), 40 kDa (200 ng μl$^{-1}$), 30 kDa (100 ng μl$^{-1}$) and 20 kDa (100 ng μl$^{-1}$). Different amounts of protein marker were separated by 12% SDS-PAGE. (*c*) Total proteins of *E. coli* (JM109; DH10B and BL21) were separated by 12% SDS-PAGE. The left gel was stained with PyMDI-Zn for 5 min, the right one was stained with CBB for 12 h, then destained overnight.

with a wide range of molecular weight (10–120 kDa). Thus, based on our data, we believe that PyMDI-Zn could be applicable as a good fluorescent gel stain.

## 3.3. Cellular location with PyMDI-Zn

Our previous data show that *E. coli* cells could be stained and lighted up with PyMDI-Zn (electronic supplementary material, figure S1), indicating the potential to apply it in cell imaging. Therefore, mammalian cells (Hela and HepG2) were incubated with PyMDI-Zn and imaged by a confocal microscopy (electronic supplementary material, figure S5B and figure 3*a*). The cells were basically green-stained, and the nucleus areas show higher brightness under laser irradiation (figure 3*a*), which is reasonable because the nucleus consists of approximately 90% proteins by dry weight. This conclusion was further confirmed by the co-staining experiment with DAPI (DNA-binding fluorescent dye for nucleus), shown in figure 3*b* and electronic supplementary material, figure S5C, and the merged image of PyMDI-Zn and DAPI, revealing that the nucleus could be stained by PyMDI-Zn (figure 3*c* and electronic supplementary material, figure S5D). Apart from this, there were some highlighted particles that appeared in the nucleus (as indicated by the arrowhead in figure 3*a* and electronic supplementary material, figure S5B), which we suspect might be nucleolus because the nucleolus consists of the high concentration of protein and could appear in the nucleus during cell-cycle progression [25–28]. Therefore, to verify this speculation, Pyronin Y, a nucleolus probe, was used to counterstain with PyMDI-Zn (as indicated by arrowhead in figure 3*d* and electronic supplementary material, figure S6B,C), the merged image of PyMDI-Zn and Pyronin Y indicated that those highlighted particles are indeed the nucleolus (figure 3*e* and electronic supplementary material, figure S6D). For live cell imaging, the biggest challenge is to maintain cell viability in the fluorescence microscopy. Therefore, the cytotoxic potential of PyDMI-Zn was investigated with Alamar Blue

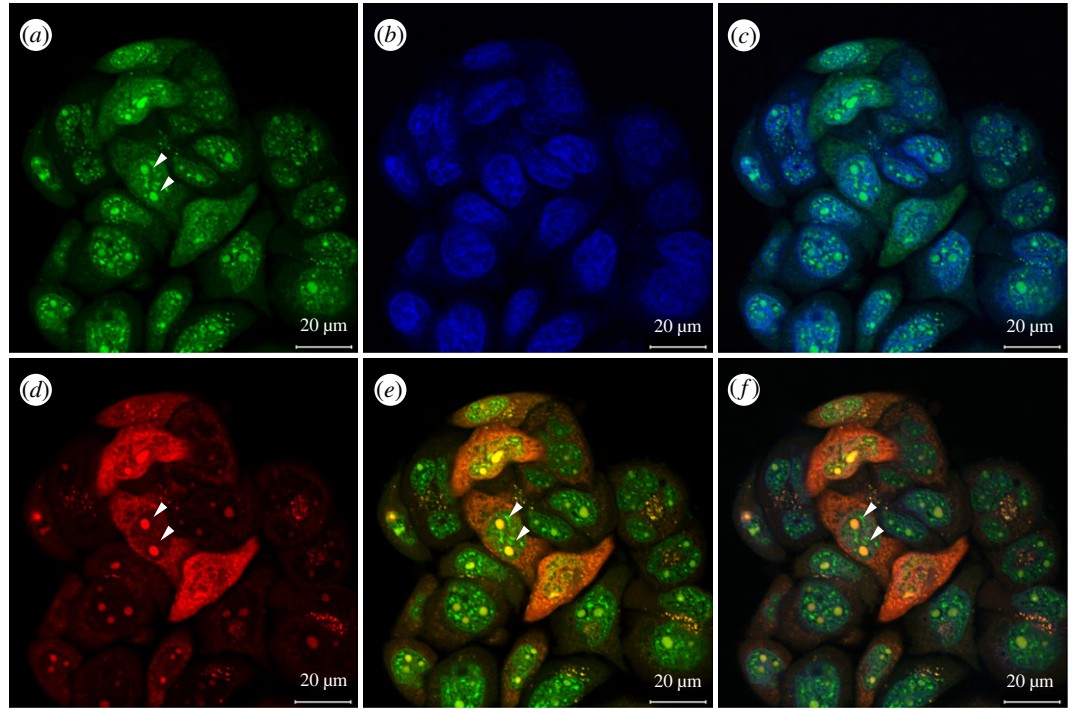

**Figure 3.** HepG2 cells were counterstained with PyMDI-Zn, DAPI and Pyronin Y. (*a*) PyMDI-Zn, (*b*) DAPI, (*c*) the merged image of PyMDI-Zn and DAPI, (*d*) Pyronin Y, (*e*) the merged image of PyMDI-Zn and Pyronin Y, (*f*) the merged image of these three dyes. Nucleus was stained with DAPI, nucleolus (white arrowhead) was stained with Pyronin Y, nucleus and nucleolus could be stained with PyMDI-Zn in the meantime from the merged picture. Scale bars 20 µm.

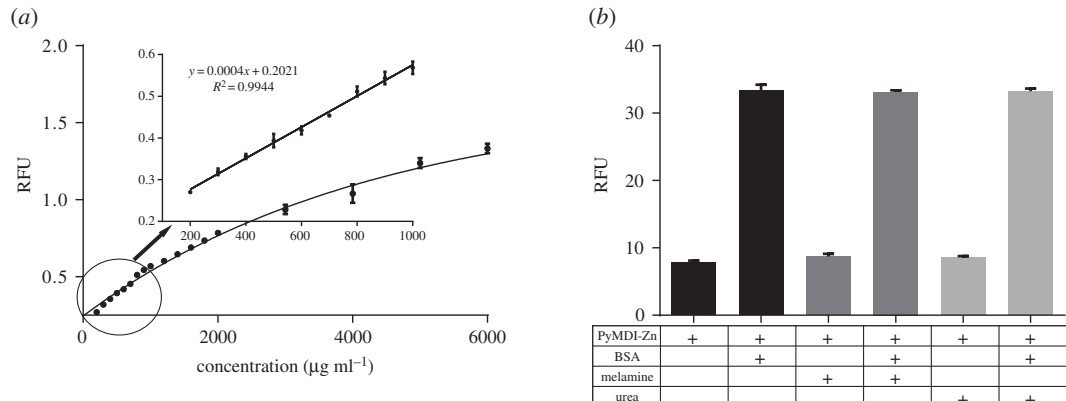

**Figure 4.** Protein quantitation with PyMDI-Zn. (*a*) Calibration plot for BSA using PyMDI-Zn. The insert reveals the sensitivity of the assay in giving a limit of detection of approximately 0.9 µg ml$^{-1}$. (*b*) Detection of contamination protein samples. Melamine or urea was added into the test solution containing protein and PyMDI-Zn respectively. The mixed samples were incubated at room temperature for about 5 min and this assay was tested at wavelength 486 nm for excitation and 520 nm for emission by Thermo Scientific Varioskan Flash.

Viability assays, showing that PyMDI-Zn would not reduce the cell viability of HepG2 as long as its concentration is less than 200 µM (electronic supplementary material, figure S7). Taken together, these results suggest that PyMDI-Zn could be applied to locate protein-rich regions and organelles in live cell imaging.

## 3.4. Protein quantitation with PyMDI-Zn

The fluorescent signals of PyMDI-Zn–protein complex increased along with the rising of protein concentration (BSA as a model protein, figure 4*a*). The emission intensities at 520 nm for PyMDI-Zn were plotted as a function of the BSA concentration, and a typical calibration graph of the response to BSA under the optimum experimental conditions was obtained. The plot indicates a good linear

relationship between the emission intensities and the BSA concentration (up to 1 mg ml$^{-1}$ of BSA, $r >$ 0.994). The detection limit was 0.9 µg ml$^{-1}$ (signal-to-noise ratio was 3.0). Methods currently used for the determination of the protein content of foodstuffs, including the Kjeldahl and Dumas Methods, depend on the determination of nitrogen [29,30]. Therefore, nitrogen-rich compounds, such as melamine and urea, have been added to correct the apparent milk protein content by fraudulent producers. As shown in figure 4b, the addition of melamine or urea did not cause the increase of the fluorescent response of PyMDI-Zn to BSA (a standard protein). In order to verify the reliability of our method in detecting real samples, we compared PyMDI-Zn-based protein quantitation with existing methods, including the Kjeldahl method and Pierce kit (electronic supplementary material, figure S8). As expected, the Kjeldahl method was easily interfered with by nitrogen-rich compounds such as melamine and urea. Although both the PyMDI-Zn method and Pierce Kit provided consistent results which were not interfered with by melamine and urea, the protein concentration measured by PyMDI-Zn probe is closer to that of the Kjeldahl method.

# 4. Conclusion

The present study demonstrates a light-up fluorophore PyMDI-Zn which could specifically bind to proteins and provide a red-shifted fluorescent emission. PyMDI-Zn can quickly respond to different proteins with a wide range of molecular weights and resist the interference of most foreign substances. By using PyMDI-Zn as a dye, the fast protein staining in SDS-PAGE could be realized in 5 min, providing great convenience. Because of the good cell penetration and low toxicity, PyMDI-Zn has been successfully applied to locate protein-rich regions or organelles in live cell imaging, and its excitation/emission maximum (488/520 nm) is the most commonly used channel on the most laser-based cell-analysis instrument. Moreover, unlike the current official method based on the analysis of nitrogen content, direct protein quantitation can be realized based on PyMDI-Zn, providing a more accurate and reliable method for determining food proteins. These results demonstrate that the fluorophore PyMDI-Zn described here is a versatile light-up probe for protein research and food quality control.

Data accessibility. The datasets supporting this article have been uploaded as part of the electronic supplementary material.

Authors' contributions. J.Z., G.C., F.D., Y.Y. and Z.T. carried out the molecular and cell laboratory works, participated in data analysis, carried out the design of the study and drafted the manuscript; J.Z., K.X., X.H., J.D., X.C. and Z.T. carried out the statistical analysis and collected the field data. J.Z. and Z.T. conceived of the study, designed the study, coordinated the study and helped draft the manuscript. All authors gave final approval for publication.

Competing interests. We declare we have no competing interests.

Funding. This work was supported by the National Natural Science Foundation of China (grant nos. 21572222, 21708037, 21877108); the Innovative Team of Sichuan Province (grant no. 2017TD0021); Chengdu Municipal Bureau of Science and Technology (grant nos. 2015-HM02-00099-SF, 2016-HM01-00371-SF); China Postdoctoral Science Foundation (grant no. 2017M612998).

Acknowledgements. The authors are grateful for the valuable comments from the editors and the reviewer, which have substantially improved the quality of their work.

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
