## [Reviewer comments · Royal Society Open Science]

Review History

RSOS-190580.R0 (Original submission)

Review form: Reviewer 1

Is the manuscript scientifically sound in its present form?

No

Are the interpretations and conclusions justified by the results?

No

Is the language acceptable?

Yes

Is it clear how to access all supporting data?

Yes

Do you have any ethical concerns with this paper?

No

Have you any concerns about statistical analyses in this paper?

No

Recommendation?

Major revision is needed (please make suggestions in comments)

Comments to the Author(s)

The authors reported a new function of an interesting compound (PyMDI-Zn). PyMDI-Zn can be used as the fluorescent dye for detecting proteins. In general, the main findings of their work are fairly interesting. However, I am afraid that the manuscript is not carefully prepared and revisions must be needed to improve the work.

The main concern about the manuscript.

As stated in the title and the whole manuscript, PyMDI-Zn can be used as staining and quantifying proteins. The works for staining proteins is sound. However, the proof for quantifying proteins is insufficient. The authors provided a calibration curve to show the linearity of the method, and a result showed that fluorescent intensities of protein-PyMDI-Zn are not influenced by urea or melamine. These works are necessary, but not enough. At least they should provide one application of this method for quantifying the protein concentration, to show that the method they proposed really works well when quantifying proteins. Samples such as milk with and without urea/melamine might be appropriate. It should be easy to carry out such experiments since they have already constructed the calibration curve. Traditional method for quantifying protein concentration may be included as control to prove the accuracy of their method. Kjeldahl or Dumas methods could be included as well to show that their method is better.

Other comments

1. the x- or y- axis titles are missing in some of the figures such as fig 1A and 1B
2. fig 1C, the authors studied the influence of proteins, DNA and RNA to the fluorescence. Have the authors excluded the influence of the sugar/polysaccharides?
3. part 4.1, change "6times" to "6 times"
4. fig 2, the same figures appeared in the supplementary materials (fig S2, S3, S4). Repeated usage of figures in this paper is not necessary. Moreover, the fig 2B and fig S3 are exactly the same image, which is not acceptable. Fig S2 and fig S4 provided CBB results for control to fig 2A and 2C. CBB staining is necessary for readers to compare the methods. Therefore, it is suggested that the fig S2-4 are removed, and CBB images in fig S2 and fig S4 are moved to fig 2.
5. fig 2B, what proteins are used in this SDS-PAGE?
6. the author stated that the proteins bands in SDS-PAGE could be seen under UV with or without washing step. Thus, it is expected that some images should be provided (at least as supplementary materials) to support the claim.
7. description about the results in table S1 is not clear to me. The table provided the concentration of many substances. What are the responses of protein-PyMDI-Zn to each substances at the given concentration? Any data?
8. there are no descriptions in the method section about how the experiments in part 4.4 are performed.

Review form: Reviewer 2 (Tian-Yun Wang)**Is the manuscript scientifically sound in its present form?**

Yes

Are the interpretations and conclusions justified by the results?

Yes

Is the language acceptable?

No

Is it clear how to access all supporting data?

Yes

Do you have any ethical concerns with this paper?

No

Have you any concerns about statistical analyses in this paper?

No

Recommendation?

Accept with minor revision (please list in comments)

Comments to the Author(s)

This paper reports a fluorescent dye, PyMDI-Zn, which could specifically bind proteins and provide a red-shifted fluorescent emission, this study is interesting and provide a useful contribution to the literature. However, some issues need to be solved before publication.

- 1) The style of this manuscript is confused, such as Figures.
- 2) The language needs to be revised and improved.
- 3) The methods should be compared with other Fluorescent Probe methods, not only CBB.

Decision letter (RSOS-190580.R0)

10-Jun-2019

Dear Dr Tang,

The editors assigned to your paper ("A General Fluorescent Light-Up Probe for Staining and Quantifying Protein") have now received comments from reviewers. We would like you to revise your paper in accordance with the referee and Associate Editor suggestions which can be found below (not including confidential reports to the Editor). Please note this decision does not guarantee eventual acceptance.

Please submit a copy of your revised paper before 03-Jul-2019. Please note that the revision deadline will expire at 00.00am on this date. If we do not hear from you within this time then it will be assumed that the paper has been withdrawn. In exceptional circumstances, extensions may be possible if agreed with the Editorial Office in advance. We do not allow multiple rounds of revision so we urge you to make every effort to fully address all of the comments at this stage. If deemed necessary by the Editors, your manuscript will be sent back to one or more of the original reviewers for assessment. If the original reviewers are not available, we may invite new reviewers.

To revise your manuscript, log into <http://mc.manuscriptcentral.com/rsos> and enter your Author Centre, where you will find your manuscript title listed under "Manuscripts with Decisions." Under "Actions," click on "Create a Revision." Your manuscript number has been

appended to denote a revision. Revise your manuscript and upload a new version through your Author Centre.

- Data accessibility

If you wish to submit your supporting data or code to Dryad (<http://datadryad.org/>), or modify your current submission to dryad, please use the following link:
<http://datadryad.org/submit?journalID=RSOS&manu=RSOS-190580>

- Competing interests

- Authors' contributions

- Acknowledgements

- Funding statement

on behalf of Professor Luning Liu (Associate Editor) and Pietro Cicuta (Subject Editor)
openscience@royalsociety.org

Associate Editor's comments (Professor Luning Liu):

Thank you for giving us the opportunity to review your manuscript for publication in Royal Society Open Science. You will see from reviewers' comments copied below that while they find your work of considerable potential interest, they have raised substantial concerns that must be addressed. In light of these comments, we cannot accept the manuscript for publication, but would be prepared to consider a revised version that addresses these concerns.

We hope you will find the referees' comments useful as you decide how to proceed. The major points raised by the Reviewers would need to be addressed if you wished to resubmit a new version of this manuscript. In particular, we agree with both referees that comparison between the present method with other quantification methods would greatly strengthen your conclusions.

Please bear in mind that we will be reluctant to approach the referees again in the absence of major revisions, and the major revision does not imply the paper will be accepted eventually. If you choose to revise your manuscript taking into account all reviewer and editor comments, please highlight all changes in the manuscript text file.

We are committed to providing a fair and constructive peer-review process. We do not generally allow multiple rounds of revision. Based on the concerns to be addressed, we would suggest 30 days to revise. If you do have any queries or concerns, you are welcome to contact me in the first instance.

Thank you again for allowing Royal Society Open Science to review your work. We look forward to hearing from you soon.

Sincerely,
Dr Luning Liu

Comments to Author:

Reviewers' Comments to Author:

Reviewer: 1

Comments to the Author(s)

The authors reported a new function of an interesting compound (PyMDI-Zn). PyMDI-Zn can be used as the fluorescent dye for detecting proteins. In general, the main findings of their work are fairly interesting. However, I am afraid that the manuscript is not carefully prepared and revisions must be needed to improve the work.

The main concern about the manuscript.

As stated in the title and the whole manuscript, PyMDI-Zn can be used as staining and quantifying proteins. The works for staining proteins is sound. However, the proof for quantifying proteins is insufficient. The authors provided a calibration curve to show the linearity of the method, and a result showed that fluorescent intensities of protein-PyMDI-Zn are not influenced by urea or melamine. These works are necessary, but not enough. At least they should provide one application of this method for quantifying the protein concentration, to show that the method they proposed really works well when quantifying proteins. Samples such as milk with and without urea/melamine might be appropriate. It should be easy to carry out such experiments since they have already constructed the calibration curve. Traditional method for quantifying protein concentration may be included as control to prove the accuracy of their method. Kjeldahl or Dumas methods could be included as well to show that their method is better.

Other comments

1. the x- or y- axis titles are missing in some of the figures such as fig 1A and 1B
2. fig 1C, the authors studied the influence of proteins, DNA and RNA to the fluorescence. Have the authors excluded the influence of the sugar/polysaccharides?
3. part 4.1, change "6times" to "6 times"
4. fig 2, the same figures appeared in the supplementary materials (fig S2, S3, S4). Repeated usage of figures in this paper is not necessary. Moreover, the fig 2B and fig S3 are exactly the same image, which is not acceptable. Fig S2 and fig S4 provided CBB results for control to fig 2A and 2C. CBB staining is necessary for readers to compare the methods. Therefore, it is suggested that the fig S2-4 are removed, and CBB images in fig S2 and fig S4 are moved to fig 2.
5. fig 2B, what proteins are used in this SDS-PAGE?
6. the author stated that the proteins bands in SDS-PAGE could be seen under UV with or without washing step. Thus, it is expected that some images should be provided (at least as supplementary materials) to support the claim.
7. description about the results in table S1 is not clear to me. The table provided the concentration of many substances. What are the responses of protein-PyMDI-Zn to each substances at the given concentration? Any data?
8. there are no descriptions in the method section about how the experiments in part 4.4 are performed.

Reviewer: 2

Comments to the Author(s)

This paper reports a fluorescent dye, PyMDI-Zn, which could

specifically bind proteins and provide a red-shifted fluorescent emission, this study is interesting and provide a useful contribution to the literature. However, some issues need to be solved before publication.

- 1) The style of this manuscript is confused, such as Figures.
- 2) The language needs to be revised and improved.
- 3) The methods should be compared with other Fluorescent Probe methods, not only CBB.

Author's Response to Decision Letter for (RSOS-190580.R0)

See Appendix A.

Decision letter (RSOS-190580.R1)

29-Jul-2019

Dear Dr Tang,

I am pleased to inform you that your manuscript entitled "A General Fluorescent Light-Up Probe for Staining and Quantifying Protein" is now accepted for publication in Royal Society Open Science.

on behalf of Professor Luning Liu (Associate Editor) and Pietro Cicuta (Subject Editor)
openscience@royalsociety.org

Appendix A

Dr. Zhuo Tang
Natural Product Center
Chengdu Institute of Biology
Chinese Academy of Sciences
Sichuan, Chengdu, 610041, P. R.
China
Fax: +86-28-85243250
E-mail: tangzhuo@cib.ac.cn
Jul. 14th, 2019

Dear editor:

Enclosed please find the manuscript (**ID: RSOS-190580**) titled “A General Fluorescent Light-Up Probe for Staining and Quantifying Protein” by Jiawei Zou, Gangyi Chen, Feng Du, Yi Yuan, Xin Huang, Juan Dong, Kexin Xie, Xin Cui, Zhuo and Zhuo Tang. This manuscript has been submitted to *Royal Society Open Science* on Apr 3th, 2019.

On behalf of my co-authors, I appreciate all reviewers very much for their professional comments and suggestions on our manuscript. Those comments are all valuable and very helpful for revising and improving our paper. We also sincerely thank the editor for allowing us to revise our manuscript. The Decision on the manuscript (**ID: RSOS-190580**) is attached and the key revisions, detailed changes and arguments can be found in the attached **Replies to the Comments by Referees**.

Best regard:

Zhuo Tang

Chengdu Institute of Biology, CAS

Reply to the Comments by Reviewer 1

We greatly appreciate the **Reviewer 1** for his/her positive comments: “*The authors reported a new function of an interesting compound (PyMDI-Zn). PyMDI-Zn can be used as the fluorescent dye for detecting protein. In general, the main finding of their work are fairly interesting.*” Meanwhile, we thank **reviewer 1** for the professional suggestions for modification of our manuscript, as well as his/her careful reading to help us find the errors in our manuscript. We have modified the paper accordingly, and our point-by-point replies to the questions are as follows:

The main concern about the manuscript

As stated in the title and the whole manuscript, PyMDI-Zn can be used as staining and quantifying proteins. The works for staining proteins is sound. However, the proof for quantifying proteins is insufficient. The authors provided a calibration curve to show the linearity of the method, and a result showed that fluorescent intensities of protein-PyMDI-Zn are not influenced by urea or melamine. These works are necessary, but not enough. At least they should provide one application of this method for quantifying the protein concentration, to show that the method they proposed really works well when quantifying proteins. Samples such as milk with and without urea/melamine might be appropriate. It should be easy to carry out such experiments since they have already constructed the calibration curve. Traditional method for quantifying protein concentration may be included as control to prove the accuracy of their method. Kjeldahl or Dumas methods could be included as well to show that their method is better.

Thank the reviewer for his/her professional question. To compare with the existing methods, we adopted the reviewer’s opinion to use milk as the sample for quantifying the protein concentration. As shown in Figure S8, we prepare three samples:

sample 1: milk powder (labeled with a protein concentration of 18% by the producer);

sample 2: milk powder with melamine to make the protein concentration up to 35% (calculated by nitrogen %);

sample 3: milk powder with urea to make the protein concentration up to 30% (calculated by nitrogen %).

We establish standard curves of quantitative methods of Pierce and PyMDI-Zn using BSA as a standard protein. The Kjeldahl method is a means of determining the nitrogen content of organic and inorganic substances, and the nitrogen content of the protein usually accounts for about 16% of its total mass. Therefore, the total protein content of the sample can be calculated by measuring the nitrogen content of the sample (assume that the nitrogen is all from the protein). And the quantitative results of three milk sample with different methods were illustrated Figures S8 as follows:

Figure S8

These data implied that the Kjeldahl method is likely to be applied by the producer of our milk sample. But this method was easily interfered by nitrogen-rich compounds such as melamine and urea. PyMDI-Zn and Pierce Kit both provided a stable data which was not interfered by melamine and urea. Moreover, the protein concentration measured by PyMDI-Zn is closer to that of the Kjeldahl method than Pierce Kit.

Question 1.

The x- or y- axis titles are missing in some of the figures such as fig 1A and 1B.

We have added x- or y-axis titles to **Figure 1A** and **Figure 1B** accordingly.

Question 2.

Fig 1C, the authors studied the influence of proteins, DNA and RNA to the fluorescence. Have the authors excluded the influence of the sugar/polysaccharides?

Thank the reviewer for his/her professional suggestion, and we have added the data of Glucose, Glycogen, and Starch into Figure 1C. There were no significant impacts to the fluorescence of protein-PyMDI-Zn by those substrates, showing the high specificity of PyMDI-Zn probe to protein.

Figure 1C

Question 3.

part 4.1, change “6times” to “6 times”.

Thank the reviewer for his/her helpful suggestions. We have changed “6times” to “6 times” in **part 4.1** accordingly.

Question 4.

fig 2, the same figures appeared in the supplementary materials (fig S2, S3, S4). Repeated usage of figures in this paper is not necessary. Moreover, the fig 2B and fig S3 are exactly the same image, which is not acceptable. Fig S2 and fig S4 provided CBB results for control to fig 2A and 2C. CBB staining is necessary for readers to compare the methods. Therefore, it is suggested that the fig S2-4 is removed, and CBB images in fig S2 and fig S4 are moved to fig 2.

We have removed the Figure S2-4 in the supplementary materials, and added the CBB images in Figure S2 and Figure S4 to Figure 2 according to the suggestion of reviewer 1, which is shown as follows.

Figure 2

Question 5.

fig 2B, what proteins are used in this SDS-PAGE.

Thank the reviewer for the careful reading. Protein samples were commercial protein markers that contain proteins of 80 kDa (100 ng/μl), 60 kDa (100 ng/μl), 40 kDa (200 ng/μl), 30 kDa (100 ng/μl), and 20 kDa (100 ng/μl). This information also has been added to the figure caption of Figure 2B.

Question 6.

the author stated that the proteins bands in SDS-PAGE could be seen under UV with or without washing step. Thus, it is expected that some images should be provided (at least as supplementary materials) to support the claim.

Thank the reviewer for the professional comment. As shown in the following Figure S2, almost the same gel-staining result was obtained with or without washing step. Therefore, all SDS-PAGE gels stained by PyMDI-Zn in our manuscript did not go through the washing step. And according to the suggestion of the reviewer, the results of experiments in Figure S2 were added into supplementary materials.

Figure S2

Question 7.

In Description about the results in table S1 is not clear to me. The table provided the concentration of many substances. What are the responses of protein-PyMDI-Zn to each substance at the given concentration? Any data?

Thank the reviewer for this professional question. The responses of protein-PyMDI-Zn to various substances were tested to investigate their interference to the protein assay (*J. AM. CHEM. SOC.* 2005, 127, 17799-17802). These reactions were carried out in the solution with 100 μ M BSA as well as 200 μ M PyMDI-Zn in presence of a different concentration of foreign substances. The fluorescence was detected at 520nm with excitation at 486 nm by Thermo Scientific Varioskan Flash. The maximum concentrations that give the perturbation of fluorescence intensity less than 10% can be obtained. We use glucose as an example to illustrate how we get the corresponding concentration (Figure S3): The fluorescence of protein-PyMDI-Zn with different concentration of glucose was recorded, and the maximum concentration of glucose the give the perturbation of fluorescence intensity less than 10% was about 20 mM.

Figure S3

Question 8.

There are no descriptions in the method section about how the experiments in part 4.4 are performed.

Thank the reviewer for the helpful suggestions. The following are the descriptions of how the experiments in part 4.4 are performed: “100µl solutions containing different amounts of protein (0 to 600 µg) with 100 µM PyMDI-Zn were incubated at room temperature for about 5 minutes. Then, the fluorescence of those samples was detected at 520nm with excitation at 486 nm by Thermo Scientific Varioskan Flash. The concentration of protein was plotted against the corresponding fluorescent intensity to obtain a standard curve. The protein concentration of the unknown samples was determined by the standard curve. Protein solutions are normally assayed in triplicate.” We have added this description in part 3.6 into the Materials and Methods section in our revised manuscript.

Reply to the Comments by Reviewer 2

We sincerely appreciate the **Reviewer 2** for his/her positive and professional comments. The **Reviewer 2** pointed out that: “*This study is interesting and provide a useful contribution to the literature.*” We thank **reviewer 2** for his/her useful suggestion to help us improve the manuscript. Following are our point-by-point answers to the comments and suggestions to our manuscript.

Question 1.

The style of this manuscript is confused, such as Figures.

Thank for the helpful suggestion of the reviewer. We have unified the style of our manuscript and supplementary materials.

Question 2.

The language needs to be revised and improved.

Thank the reviewer for his/her careful reading. Our manuscript has been edited by a native English speaker. The manuscript and Supplementary material with correction traces are submitted too.

Question 3.

The methods should be compared with other fluorescent probe methods, not only CBB.

Thank the reviewer for his professional suggestion. SYPRO Orange is one of the most commonly used fluorescent dye, and we chose it for the comparison experiment because it is a sensitive, ready-to-use fluorescent gel stain for proteins. After staining for typically 30-60 minutes in the dye solution, gels can be photographed after a quick rinse in 7.5% acetic acid. Protein samples were commercial protein markers, which contain proteins of 120 kDa (100ng/μl), 100 kDa (100ng/μl), 80 kDa (100ng/μl), 60 kDa (100ng/μl), 50 kDa (200 ng/μl), 40 kDa (100ng/μl), 30 kDa (100ng/μl), and 20 kDa (100ng/μl). Different amounts of protein marker were separated by 12% SDS-PAGE, and then these gels were stained with PyMDI-Zn and SYPRO Orange respectively.

Figure S4

As shown in Figure S4, SYPRO Orange afforded a lower background and a higher fluorescent response to protein bands (50-120 kDa) than PyMDI-Zn on the gel. But, as for the proteins with lower molecular weight (20, 30 and 40 kDa), PyMDI-Zn gave better staining results. And according to the suggestion of the reviewer, Figure S3 have been added into supplementary materials.

Reviewer: 1

Comments to the Author(s)

The authors reported a new function of an interesting compound (PyMDI-Zn). PyMDI-Zn can be used as the fluorescent dye for detecting proteins. In general, the main findings of their work are fairly interesting. However, I am afraid that the manuscript is not carefully prepared and revisions must be needed to improve the work.

The main concern about the manuscript.

As stated in the title and the whole manuscript, PyMDI-Zn can be used as staining and quantifying proteins. The works for staining proteins is sound. However, the proof for quantifying proteins is insufficient. The authors provided a calibration curve to show the linearity of the method, and a result showed that fluorescent intensities of protein-PyMDI-Zn are not influenced by urea or melamine. These works are necessary, but not enough. At least they should provide one application of this method for quantifying the protein concentration, to show that the method they proposed really works well when quantifying proteins. Samples such as milk with and without urea/melamine might be appropriate. It should be easy to carry out such experiments since they have already constructed the calibration curve. Traditional method for quantifying protein concentration may be included as control to prove the accuracy of their method. Kjeldahl or Dumas methods could be included as well to show that their method is better.

Other comments

- 1. the x- or y- axis titles are missing in some of the figures such as fig 1A and 1B*
- 2. fig 1C, the authors studied the influence of proteins, DNA and RNA to the fluorescence. Have the authors excluded the influence of the sugar/polysaccharides?*
- 3. part 4.1, change "6times" to "6 times"*
- 4. fig 2, the same figures appeared in the supplementary materials (fig S2, S3, S4). Repeated usage of figures in this paper is not necessary. Moreover, the fig 2B and fig S3 are exactly the same image, which is not acceptable. Fig S2 and fig S4 provided CBB results for control to fig 2A and 2C. CBB staining is necessary for readers to compare the methods. Therefore, it is suggested that the fig S2-4 are removed, and CBB images in fig S2 and fig S4 are moved to fig 2.*
- 5. fig 2B, what proteins are used in this SDS-PAGE?*
- 6. the author stated that the proteins bands in SDS-PAGE could be seen under UV with or without washing step. Thus, it is expected that some images should be provided (at least as supplementary materials) to support the claim.*
- 7. description about the results in table S1 is not clear to me. The table provided the concentration of many substances. What are the responses of protein-PyMDI-Zn to each substances at the given concentration? Any data?*
- 8. there are no descriptions in the method section about how the experiments in part 4.4 are performed.*

Reviewer: 2

Comments to the Author(s)

This paper reports a fluorescent dye, PyMDI-Zn, which could specifically bind proteins and provide a red-shifted fluorescent emission, this study is interesting and provide a useful contribution to the literature. However, some issues need to be solved before publication.

1) *The style of this manuscript is confused, such as Figures.*

2) *The language needs to be revised and improved.*

3) *The methods should be compared with other Fluorescent Probe methods, not only CBB.*